# Comparison of Usability between Gear Shifters with Varied Visual and Haptic Patterns and Complexities

**Sanna Lohilahti Bladfält \***, **Camilla Grane and Peter Bengtsson**

Department of Business Administration, Technology and Social Sciences, Luleå University of Technology, 971 87 Luleå, Sweden; camilla.grane@ltu.se (C.G.); peter.bengtsson@ltu.se (P.B.)

**\*** Correspondence: sanna.lohilahti.bladfalt@ltu.se; Tel.: +46-920-49-33-86

**Abstract:** Shift-by-wire technology enables more options concerning the design, placement and functions of gear shifters compared to traditional gear shifters with manual transmission. These variations can impact usability and driver performance. There is a lack of research regarding the potential advantages and disadvantages of different types of gear shifters. The purpose of this study was to investigate the efficiency and subjective ease-of-use of mono- and polystable joystick gear shifter types at different complexity levels and with full or limited visibility. An experimental study with 36 participants was conducted. The results showed that monostable joysticks, especially those with an I/J-shape, were overall less efficient and easy to use than polystable joysticks. The highest complexity level clearly affected the efficiency for the monostable joystick with an I/J-shape (mono I/J) compared with the other gear shifter types. The monostable joystick with an I/J-shape (mono I/J) was also most affected by reduced visibility at the highest level of complexity, indicating that it was more prone to causing users to take their eyes off the road.

**Keywords:** usability; complexity; shift-by-wire; gear shifter

---

## 1. Introduction

In 1968, the US National Highway Traffic Safety Administration (NHTSA) alleged that diversity among automatic gear shifters could potentially negatively affect traffic safety [1]. Thus, the NHTSA introduced legislation to prevent errors by standardizing the well-established PRND (Park, Reverse, Neutral and Drive) shift lever sequence. The NHTSA was particularly concerned about shifter errors in association with unfamiliar vehicles and infrequent drivers. This reasoning was in accordance with contemporary research emphasizing that the infrequently used functions in a vehicle should have good learnability and memorability [2].

The linkage between the gear shifter and transmission has traditionally been mechanical, with large and heavy components such as a Bowden cable [3]. Today, the mechanical components can be removed and replaced by electronic shift-by-wire technology, which is becoming increasingly common in modern cars with automatic transmissions [4]. Shift-by-wire gear shifters are often monostable, meaning that the gear shifter will return to one position when the driver releases it. Gear shifters with mechanical transmissions are instead polystable, and the gear shifter will stay in the position to which the driver moves it. Shift-by-wire technology enables new options concerning the design and function of the gear shifter as well as the overall interior design and functionality of the car since the gear shifter is no longer restricted by a mechanical connection [3]. However, as electronic by-wire technology offers car designers increased degrees of freedom, the drivers might face issues regarding usability. A monostable gear shifter lacks the typical spatial haptic feedback when moving the shifter into park, drive or reverse. Monostable shifting requires the driver to devote time to confirm the desired gear position on a display. In a worst-case scenario, this could have fatal consequences.

The NHTSA concluded that the operation of a monostable shifter is not intuitive, which increases the potential for unintended gear selection [5]. Studies into both manual and automatic gear shifting suggest that drivers might fail to notice traffic information and also make gear-shift errors when gear shifting in attention-demanding or stressful situations [6,7]. A tragic accident caused by a miss-shift occurred in 2015, in which a driver tried to reverse her car that was standing too close a railway crossing. Instead of reversing, the driver drove the car forward; the driver and five train passengers were killed [8]. Based on consumer reports, there have been numerous incidents related to electronic gear shifters [9–12]. One driver described a situation in which he exited the car, believing the gear was in "Park", and the car began to reverse out of the driveway. Another driver made a similar mistake and reversed into a car [10]. Due to several hundred numbers of unintended vehicle rollaway incidents and alleged crashes associated with a 2014–2015 specific car model, the NHTSA opened an investigation into monostable gear shifters [13]. The problems identified involved more than one model and manufacturer; consequently, the car industry recalled more than a million vehicles on account of traffic safety. The cars were equipped with electronic monostable gear shifters, conveying the selected gear by indicator lights instead of the gear shifter position [13]. According to [14–17], haptics can be used to relieve the visual load and make the vehicle interface more intuitive to use. Findings like these emphasize the importance of haptics when handling secondary tasks such as gear shifting in cars.

Consequently, the transition towards monostable gear shifters raises several questions from a traffic safety perspective. There are several studies available on gear shifters [18–20]; however, the implications for usability have not been thoroughly explored. Studies comparing and evaluating monostable and polystable gear shifters are lacking [21,22]. Nevertheless, developers and producers of monostable gear shifters have to continuously make design solutions related to usability and safety issues with the new technology. A study on monostable gear shifters in new vehicles [21] concluded that highly innovative gear shifter designs might confuse drivers and have a negative impact on usability. Furthermore, there seemed to be differences in usability between various monostable designs such as a joystick, stalk, rotary or button shifters.

The aim of this paper is to evaluate the usability aspects of monostable versus polystable joystick gear shifters at different complexity levels, with or without visual feedback. Since some cognitive functions, such as the speed of processing information, seems to begin to decline for adults at the age of 20 or 30 with an acceleration of decline at the age of 50 [23,24], only younger drivers under the age of 35 were chosen as participants in this study. More specifically, three hypotheses were tested.

**Hypothesis 1.** *There is a difference in efficiency and ease-of-use between shifter types, i.e., monostable and polystable joystick gear shifters.*

**Hypothesis 2.** *There is a difference in efficiency and ease-of-use between joystick gear shifters at different complexity levels, i.e., with a different number of choices.*

**Hypothesis 3.** *There is a difference in efficiency and ease-of-use between joystick gear shifters depending on visual feedback.*

## 2. Materials and Methods

### 2.1. Participants

The study comprised 36 participants (29 M; 7 W) with driver's licenses who were between 19 and 31 years old (M = 22.8; SD = 2.7). All participants drove on a regular basis and had—except for two participants—experience of driving a car with automatic transmission. The participants were students at Luleå University of Technology and received information regarding the study by email. If the students had driver's licenses, they could apply to participate by choosing an appropriate time and date.

Participation was voluntary, and the students were not in any position of dependency with the project team. Participants received detailed verbal, as well as written, information about their participation in the study. They were informed about the plan and purpose of the study, the methods to be used, the parties responsible for the study, that participation was voluntary and that they could interrupt and discontinue their participation at any time. The participants received information stating that their anonymity would be maintained and that the study was not measuring individual performance; the study solely measured the performance of the gear shifters. After this, the participants gave their written and verbal consent before actual participation. The study followed Swedish laws regarding conducting experimental research, [25] as well as research rules and guidelines [26]. The participants received a cinema ticket voucher to the value of 135 Swedish crowns.

## 2.2. Equipment

A gear shifter prototype with exchangeable movement patterns was used in the study (Figure 1). The gear shifter prototype was developed by Kongsberg Automotive in the form of a joystick, one of the most common gear shifter types in production today. The structural components were made of aluminum with a gimbal joint equipped with ball bearings. The shift force was generated by a spring-loaded plunger acting against an index landscape; the shift pattern and number of shift positions could be altered by exchanging the index component. The panel and knob were made out of 3D-printed, painted polyamide (SLS). The movement patterns were constructed based on vehicle standards regarding movement resistance and the distance between shift positions. The gear shifter visibility was regulated through a blanket fixed on a stick that was lowered over the participant's right shoulder, which limited their vision but not their hand movement between the steering wheel and gear shifter. The participants were seated in a car seat in front of a desktop driving simulator. The gear shifter was placed on the right-hand side of the participant, at a height and distance that could be assumed to be a common distance in a traditional passenger car with a joystick gear shifter, i.e., approximately 33 cm from the backrest of the seat and 12 cm above the sitting surface of the seat. However, the car seat could be moved so that the gear shifter was at a comfortable distance for the participant, although the height of the seat could not be altered. The desktop driving simulator consisted of a 24-inch computer screen and a Logitech steering wheel; no pedals were used. Task instructions regarding gear shifts were presented to the participants on a 21-inch computer screen, both as capital letters on the computer screen as well as verbal instructions through the speakers in front of the participant. The task instruction screen was placed to the left of the desktop simulator screen; see Figure 2 for the experimental set-up. The participant's interaction with the gear shifters was recorded with four webcams using vMix software (Figure 3). The test leader saw the videos captured by the four webcams in real-time during the experiment while sitting on the right-hand side of the participant. The test leader screen showing the webcam videos was hidden from the participant's sight.

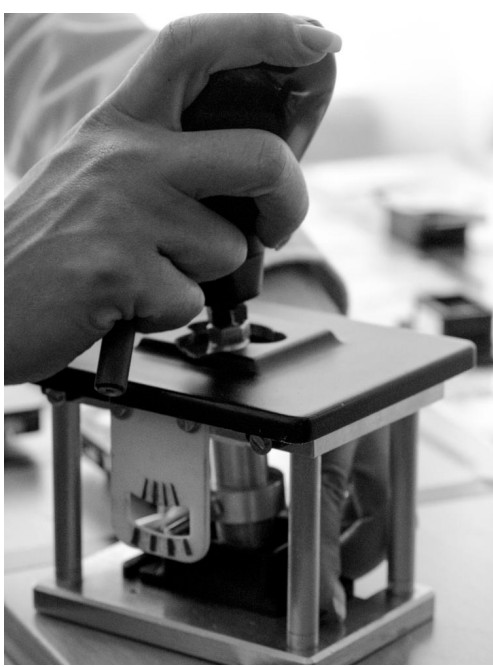

**Figure 1.** The gear shifter prototype with exchangeable movement pattern plates.

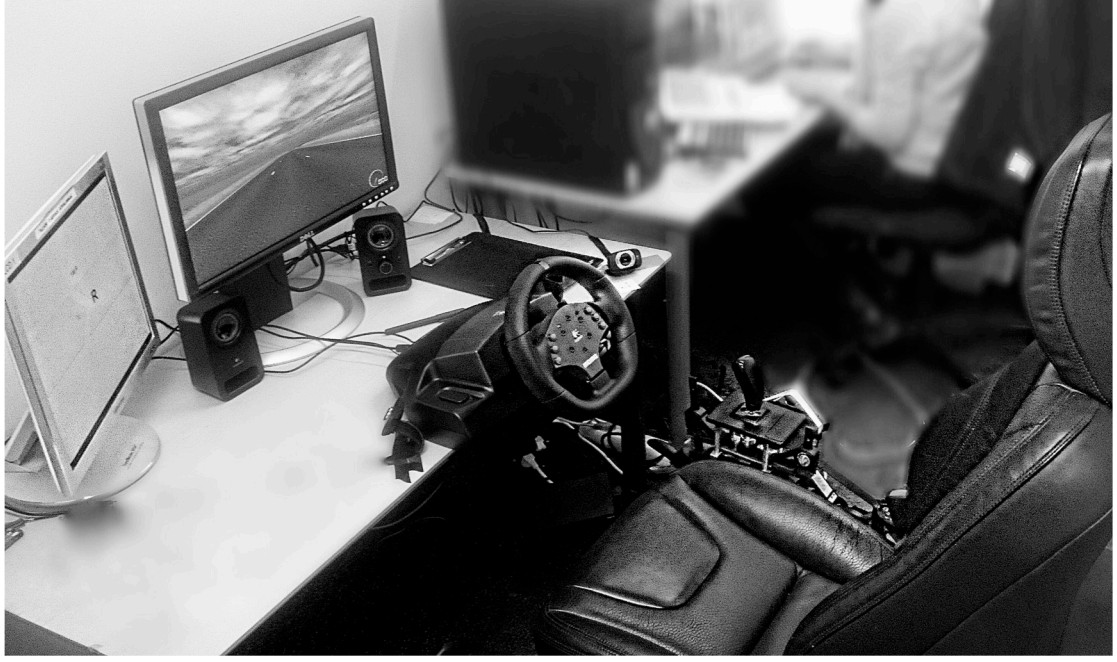

**Figure 2.** The experimental set-up. The test leader was sitting on the right-hand side of the set-up.

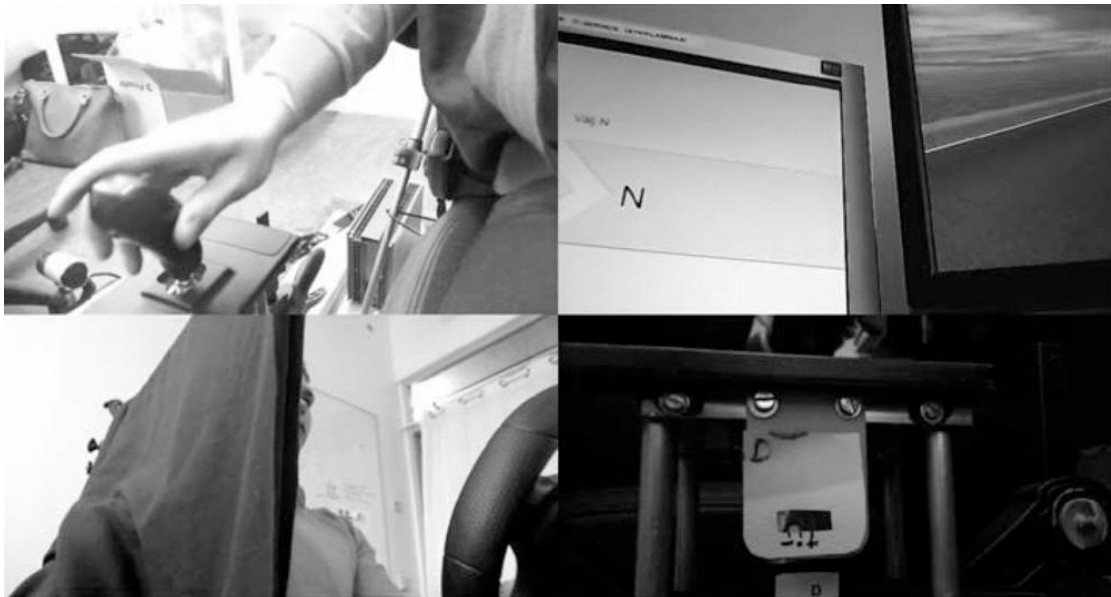

**Figure 3.** Four channel video recorded with vMix. The upper-left camera recorded the gear shifter interaction, the upper-right camera recorded the gear shifting task instructions and part of the driving scene, the lower-left camera recorded the participant (and shows the blanket used to regulate visibility) and the lower-right camera recorded the movement of the gear shifter pin in the feedback pattern plate (below joystick area).

### 2.3. Experimental Conditions

Three experimental conditions were varied in the study: shifter type, visibility state and complexity level.

### 2.3.1. Shifter Types

Four different gear shifter types were compared at three complexity levels (Figure 4). All shifters had a joystick shape (Figure 1) but differed in their movement pattern. Two gear shifter types had a monostable pattern (mono) with only one or two fixed positions, i.e., the gear shifter returned back to a central position after a gear shift. Two gear shifter types had a polystable pattern (poly); i.e., the gear shifter stayed in the selected position. The shifter types also varied in the shape of their movement pattern, which was either J-shaped, with all shift states but one presented in a vertical row and with one state positioned to the lower left, or I-shaped, with all shift states presented in a vertical row. One of the monostable shift types had a toggle function; i.e., it could only be moved one step upwards and downwards irrespective of the number of shift states available. If the user wanted to go two steps up with the monostable toggle shifter, the user had to repeat the upwards movement twice. In summary, the shifter types compared in the study had a monostable shift pattern with an I or J shape (depending on the complexity level; mono I/J), a monostable shift pattern with a toggle function (mono T), a polystable shift pattern with a J-shape (poly J) and a polystable shift pattern with an I-shape (poly I). The pattern shape of the mono I/J shifter resembled the shape available in production cars, resulting in an I-shape at a low complexity level and a J-shape at higher complexity levels. The movement patterns are illustrated in Figure 4.

**Figure 4.** Illustration of the four shifter type movement patterns at the three different levels of complexity. Stable positions are illustrated as filled circles and unstable positions as unfilled circles. Next to the movement patterns, in the lower-right corner, is the visual presentation of the shift state positions presented. Straight vertical lined patterns are named I and straight vertical lined patterns with one state positioned to the lower-left are named J. The monostable joystick with the toggle function is named mono T.

### 2.3.2. Visibility State

The gear shifters were tested under two visibility conditions: visible or non-visible. The non-visible state was added to test the possibility of changing shift state without looking away from the road. In the visible state, both the gear shifter and an instruction note with the shift state positions were visible for the participants. The notes with shift state patterns were placed next to the computer screen displaying the driving scene, on the same side as the gear shifter (the participants' right side). The size of the note was H8 × W4 (cm) with black capital letters on white paper. In the non-visible state, the instruction note was removed and a blanket was used to reduce the visibility of the gear shifter (Figure 3, in lower-left corner).

### 2.3.3. Complexity Level

The shifter types were tested at three complexity levels, as presented in Figure 3. At the lowest complexity level, the participants had to select a shift state among three choices: Reverse (R), Neutral (N) or Drive (D). At the medium complexity level, the participants had four choices: R, N, D and Manual (M). Finally, at the highest complexity level, the participants had five choices: Park (P), R, N, D and M. The M state had no additional choices. Hence, the participants could not choose to increase or decrease gears manually. Furthermore, the P-function had no additional locking features; thus, the participants could select it and deselect it without pressing the P-lock button or pressing down the brake pedal.

*2.4. Experimental Design*

The study had a mixed design since a full within-subject design would have been too comprehensive for the participants. Two experimental conditions, shifter type and visibility state, had a within-subject design. The shifter type order was counterbalanced between the participants, based on a Latin square. The visibility state had a fixed order, with the visible state first and the non-visible state presented second. The non-visible state was therefore always preceded by a learning phase. The third experimental condition, complexity level, was compared with a between-groups design with 12 participants in each group. The participants were randomly assigned to one of the three complexity levels.

2.4.1. Gear Shifting Task

The aim of this work was to study the performance of the gear shifters while participants performed a distracting task (using a driving simulator program) and frequently shifted gears. The driving performance itself was not measured. The instructions were presented with PowerPoint slides on the task instruction screen (Figure 3, in upper-right corner). The PowerPoint slides were presented as both visual images and pre-recorded voice directions, such as "Select R", in Swedish. New shifting tasks were presented every 17 seconds using timed slide shifts. Nine shift instructions were given per driving session and there were two driving sessions per gear shifter type. The first driving session had full visibility and the second had limited visibility (the non-visible state). The order of shift instructions (i.e., shift tasks) varied between driving sessions in a counterbalanced order with the aim of reducing learning effects. All four shifter types used the different shift order sets equally often. All possible shift movements were obviously not covered in each driving session, and the shift orders were therefore selected carefully with respect to having mid-positions and end-positions as targets, and the number of steps to reach a target, evenly represented.

2.4.2. Distraction Task

The gear shifters were not linked to the lane change test (LCT) [27], version 1.2. Gear shifts did not change the spatial perception of the car in the LCT. Since a driver does not normally shift gear while driving a car with automatic transmission, the LCT was used as a way of adding distraction. It was selected as a distraction task since it demands high visual focus towards the road and allows a high level of control and consistency among participants and conditions. In the task, the participants steered a fictitious car on a three-lane highway road and changed lanes when directed to by signs. A driving session lasted for about three minutes. The driving speed was set to 60 km/h. The participants were asked to hold both hands on the steering wheel between gear shifts.

2.4.3. Procedure

All participants were given the same instructions throughout the experimental session, but the order of the instructions varied due to the counterbalancing of gear shifter types. The instructions were read aloud by the experimenter from a manuscript. During the first part of the session, the participant received information regarding anonymity and the possibility of ending the session at any time. A confirmation letter was signed and the participant filled in a questionnaire regarding background data (age, gender, years with driving license and similar). In the second part of the session, the participant became acquainted with the secondary task. They drove the first track in the LCT without any additional tasks, i.e., without interaction with a gear shifter. In the third part of the session, the participants became acquainted with the gear shifting task, i.e., their first gear shifter type (mono I/J, mono T, poly J, or poly I). The experimenter explained how to make shifts with the particular shifter type, and the participant was asked to move the shifter accordingly. In order to verify a basic understanding of the particular gear shifter, the participants had to perform a gear shift test in which they had to successfully perform 10 correct gear shifts in a row (without a time limit) before proceeding

with the experiment. In the experimental test, the participants drove a second track in the LCT while simultaneously making gear shifts on demand. In the LCT, the order of the signs and the distance between the signs varied between tracks. During the three-minute drive, there were 12 signs to follow and 10 shifting tasks. The experimenter verified the gear shifts in real time. If an error was made, the test leader informed the participant about the error and the error was corrected by the participant. All gear shifters were tested with full visibility and with limited visibility. These driving sessions were completed in succession. When the first driving session was completed, a blanket was lowered over the participant's shoulder and the participant drove a second session with the same gear shifter. This procedure lasted for three minutes per gear shifter and the procedure was done with all gear shifters at the particular complexity level of the participant. After each driving session, the participants were interviewed and answered the technology acceptance model (TAM) [28] questionnaire, with questions regarding the perceived ease of use. The gear shifters were tested in an order organized according to a Latin square. In total, an experimental session lasted up to 90 minutes; see the experimental procedure for each participant in Figure 5.

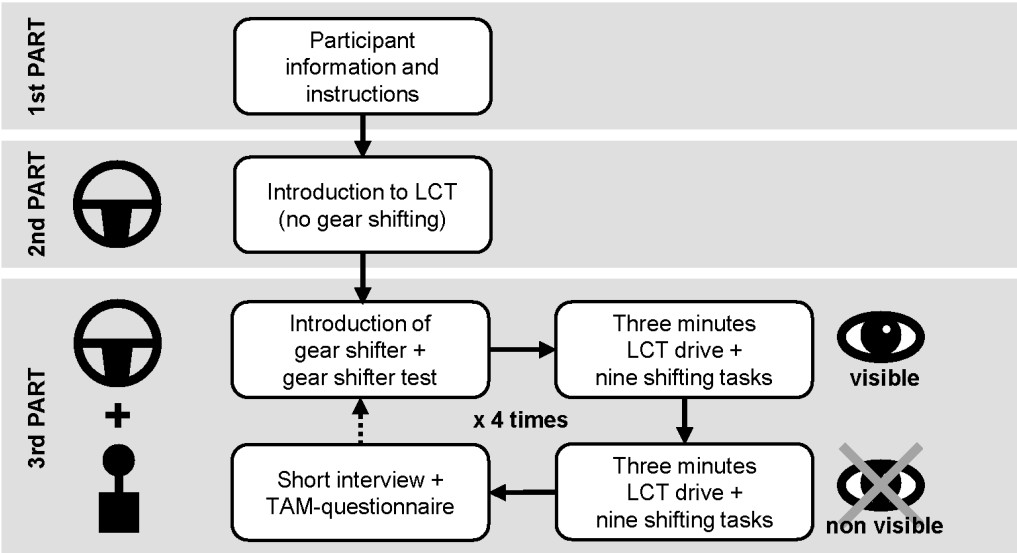

**Figure 5.** Experimental procedure from the beginning to the end. Each participant repeated the procedure once with each of the four gear shifters. LCT: lane change test.

## 2.5. Measures

### 2.5.1. Efficiency (Task Completion Time)

Usability was measured as efficiency according to ISO 9241-210 [29]. Efficiency is described as follows in ISO 9241-210: "Resources expended in relation to the accuracy and completeness with which users achieve goals". In this study, efficiency was measured as task completion time (TCT). Task completion time was measured as the time that passed from when the participant received the verbal task from the instruction computer until the participant completed the task, i.e., until they moved the gear shifter to the correct gear shifter position. A short task completion time indicated higher efficiency.

### 2.5.2. Subjective Ease of Use

Subjective evaluations of the shifters were measured by the perceived ease of use (PEOU) part of TAM [28]. High scores in the ease of use questionnaire indicated high usability by subjective ratings.

*2.6. Analysis*

The objective efficiency data (task completion time) were analyzed with a mixed model design ANOVA with two within-subject variables and one between-subject variable. Shifter types and visibility were within-subject variables, and complexity level was the between-subject variable. The subjective ease-of-use data were analyzed with a mixed model design ANOVA with one within-subject variable (shifter type) and one between-subject variable (complexity level). The subjective rating was made once per shifter, after the completion of the task in a visible and non-visible state. The analyses were calculated with the general linear model and repeated measures ANOVA with the between-subject factor, in the statistical analysis program SPSS. Post hoc analyses were made with Tukey's HSD (Honestly Significant Difference test) for the between-subject factor and a Bonferroni correction was used for the within-subject factors and interaction effects. The significance level was set to 0.05.

## 3. Results

*3.1. Results Regarding Efficiency (Task Completion Time)*

The mean task completion times and standard deviations for each shifter type, in visible and non-visible states, for each complexity level are presented in Table 1. Mauchly's test of sphericity and Levene's test of error variances were not violated.

**Table 1.** Mean results and standard deviations.

| Shifter Type | Complexity Level | Time (Visible) | | Time (Non-Visible) | | Ease of Use | |
|---|---|---|---|---|---|---|---|
| | | M | SD | M | SD | M | SD |
| Mono I/J | 3 choices | 2.25 | 0.40 | 2.27 | 0.56 | 4.91 | 0.99 |
| | 4 choices | 2.73 | 0.79 | 2.71 | 0.73 | 4.76 | 0.89 |
| | 5 choices | 3.32 | 0.76 | 3.89 | 0.93 | 3.88 | 1.28 |
| Mono T | 3 choices | 2.12 | 0.38 | 2.08 | 0.38 | 5.76 | 1.04 |
| | 4 choices | 2.63 | 0.77 | 2.59 | 0.72 | 5.61 | 0.81 |
| | 5 choices | 3.00 | 0.81 | 2.90 | 0.55 | 5.48 | 1.23 |
| Poly J | 3 choices | 1.84 | 0.39 | 1.83 | 0.39 | 6.42 | 0.52 |
| | 4 choices | 2.46 | 0.79 | 2.19 | 0.58 | 5.89 | 0.89 |
| | 5 choices | 2.43 | 0.54 | 2.29 | 0.53 | 5.88 | 0.83 |
| Poly I | 3 choices | 1.95 | 0.47 | 1.79 | 0.36 | 6.08 | 1.05 |
| | 4 choices | 2.28 | 0.49 | 2.30 | 0.51 | 5.28 | 1.23 |
| | 5 choices | 2.58 | 0.58 | 2.69 | 0.50 | 4.79 | 1.16 |

Note: M = mean; SD = standard deviation.

### 3.1.1. Shifter Types

There was a significant effect of shifter type on efficiency ($F(3,99) = 38.42$, $p = 0.000$). The pairwise comparisons revealed that the monostable shifter with an I/J-shape (mono I/J) demanded longer task completion times than the monostable shifter with a toggle (mono T; $p = 0.004$) or the polystable shifters with a J-shape ($p = 0.000$) or an I-shape ($p = 0.000$). The monostable shifter with a toggle function demanded longer task completion times than the two polystable shifters with a J-shape ($p = 0.000$) and I-shape ($p = 0.000$). No difference in efficiency was found between the two polystable shifter types with a J- or I-shape ($p = 0.589$).

### 3.1.2. Visibility State

No significant difference of efficiency was found for the visibility variable alone (F(1,33) = 0.015, *p* = 0.902). However, an interaction effect was found (see Section 3.1.4 Interaction Effects).

### 3.1.3. Complexity Levels

There was a significant effect of complexity level on efficiency (F(2,33) = 9.13, *p* = 0.001). The post hoc test revealed significant shorter task completion times at the lowest complexity level with three choices (RND) compared to the highest complexity level with five choices (PRNDM; *p* = 0.000). No significant difference was found between the complexity levels with three or four choices (*p* = 0.069) or between four and five choices (*p* = 0.138).

### 3.1.4. Interaction Effects

An interaction effect was found for the variables of shifter type and complexity level (F(6,99) = 5.44, *p* = 0.000). Figure 6 indicates an increased differentiation between shifter types at the highest complexity level with five choices (PRNDM). An interaction effect was also found for the variables of shifter type and visibility state (F(3,99) = 4.70, *p* = 0.004). Figure 7 shows that the mean task completion time decreased or was unchanged for three shifter types (mono T, poly J and poly I) in the second trial when the visibility was removed. The monostable shifter with an I/J-shape (mono I/J) showed a different pattern with an increased mean task completion time in the second trial when visibility was reduced. Finally, an interaction effect was also found for the three variables of shifter type, visibility state and complexity level (F(6,99) = 2.76, *p* = 0.016). Figure 8 shows an increased differentiation between the shifter types in the non-visible state (unfilled circles) at the highest complexity level. No interaction effect was found between visibility state and complexity level (F(2,33) = 1.59, *p* = 0.218).

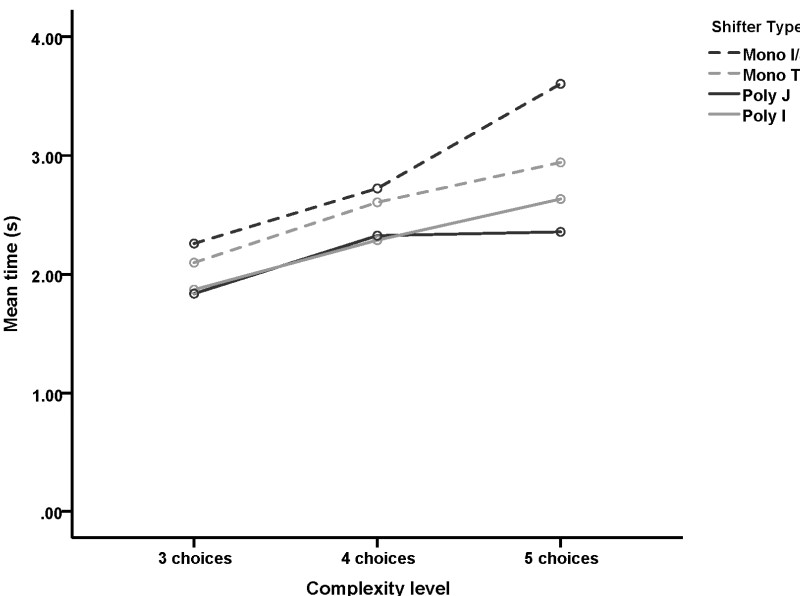

**Figure 6.** Mean task completion time for each shifter type over three levels of complexity.

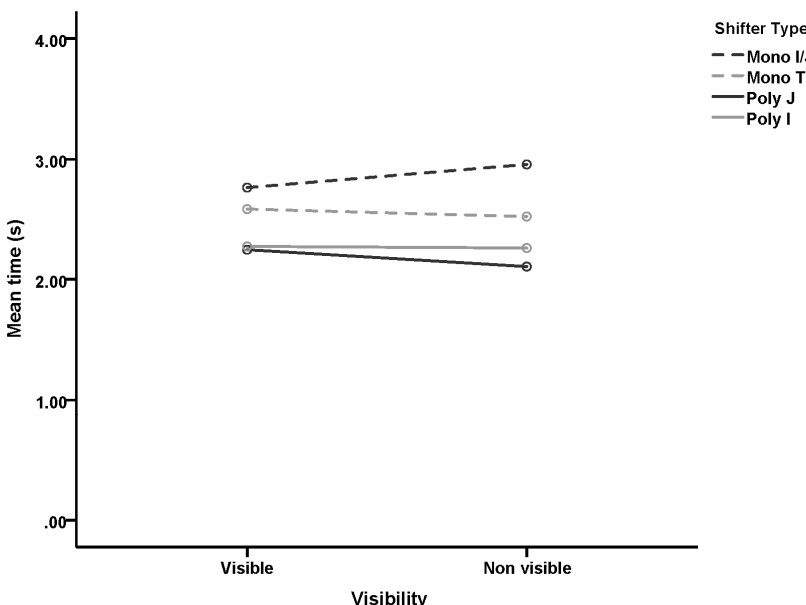

**Figure 7.** Mean task completion time for each shifter type in the two visibility states (visible and non-visible).

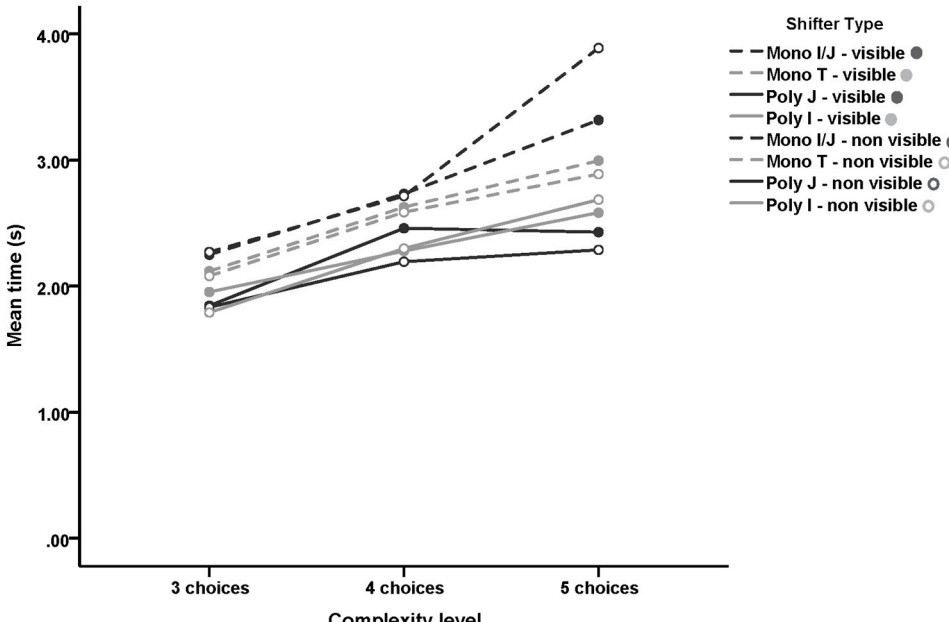

**Figure 8.** Mean task completion time for each shifter type, in both visible and non-visible states, over the three complexity levels.

*3.2. Results Regarding Subjective Ease of Use*

Figure 9 shows the subjective ease of use for each shifter type over the three complexity levels. The mean ratings and standard deviations are presented in Table 1.

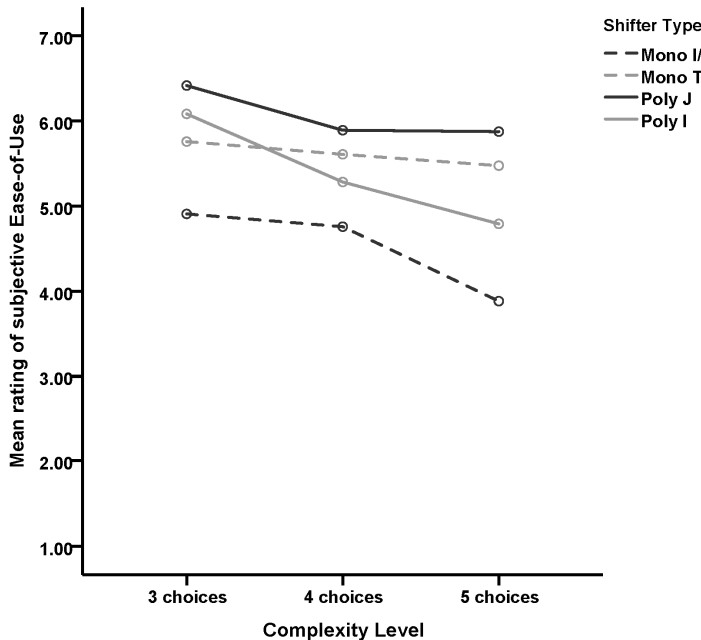

**Figure 9.** Mean rating of subjective ease of use for each shifter type and complexity level. Please note, a high rating means the shifter type was easy to use.

### 3.2.1. Shifter Types

A significant effect of subjective ease of use was found for the shifter types ($F_{(3,99)} = 16.05$, $p = 0.000$). The monostable shifter type with an I/J shape was rated as more difficult to use than all other shifter types: monostable with toggle function ($p = 0.001$), polystable with J-shape ($p = 0.000$), and polystable with I-shape ($p = 0.008$). A significant difference was also found between the two polystable shifters. The J-shaped polystable shifter was rated as easier to use than the I-shaped shifter ($p = 0.004$).

### 3.2.2. Complexity Level

A significant effect was also found for ease of use for the different complexity levels ($F_{(2,33)} = 5.79$, $p = 0.007$). The complexity level with five choices (PRNDM) was rated as more difficult to use than the level with three choices (RND; $p = 0.005$).

### 3.2.3. Interaction Effects

No interaction effect was found for shifter types and complexity levels ($F_{(6,99)} = 0.97$, $p = 0.449$).

## 4. Discussion

The results confirm Hypothesis 1; i.e., there was a difference in efficiency (task completion time) and subjective ease of use between joystick shifter types. The monostable I/J was less efficient and rated lower on ease-of-use than the other three shifters. In addition, the monostable T was less efficient than the two polystable shifters. Furthermore, the polystable I was rated lower in terms of subjective ease of use than the polystable J shifter. The outcome validates the conclusions drawn by NHTSA [5], i.e., that the monostable shifter is not intuitive, presumably due to its poor tactile and visual feedback to the driver. It is also in line with the results of Bladfält et al. [21] regarding the low usability of monostable gear shifters in vehicles. Mono I/J shifters have several positions in a row and no haptic feedback, such as a stop. Mono T has a more distinct haptic feedback, featuring stops, which prevents the shifter from sliding past the intended state. It could be that the risk of sliding past an intended state with mono I/J made it more difficult to handle than mono T. Furthermore, the driver needed to focus on feeling when a shift between positions was made as well as keeping the shifter pattern in

mind. The stopping sensation with mono T resembled the haptic feeling of the polystable gear shifters. The stopping sensation could be an advantage, which agrees with the results of [7], studying drivers' gear shifting behaviors and errors. The results showed that drivers shifted into park with an excessive use of force in the hand movement. There was a great deal of reliance on the actual kinesthetic feedback of the gear shifter when determining when the park position was reached. Interestingly, drivers who failed to shift gear or performed erroneous shifts did not pay attention to the illuminated indicators in the car to confirm the gear state [7].

The results also confirm Hypothesis 2; i.e., there was a difference in efficiency and subjective ease of use between joystick gear shifters at different complexity levels. As anticipated, the complexity level with five choices—a standard PRNDM shift sequence—was less efficient and rated as lower on ease of use than the level with three choices—a reduced RND shift sequence. The conclusion was trivial. However, interestingly, the interaction effect between shifter type and complexity displayed an increased difference in efficiency between the mono I/J and the other three shifter types along with higher complexity. Thus, the mono I/J was more negatively affected by an increased complexity level than the other shifters. It should be noted that high complexity in this study denotes a standard PRNDM shift sequence; this implies that the difference in usability between monostable and polystable gear shifters might be more noticeable in practice than suggested in this paper. Furthermore, with increased complexity, the poly I was less efficient then the poly J. It could be considered whether the J-shape made the straight-lined shifter pattern easier to use, by shortening the vertical movement, making it more difficult to slide past an intended state. It could also be hypothesized that the J-shape made the pattern easier to remember since it was divided into two movement directions instead of one, providing additional cues with a second fixed position. One of the participants described the polystable joystick with an I-shape (Poly I) as follows: "When I return to M, it's easier if there are different positions, this was only a long row so it was harder in that way . . . I kept saying PRNMD in my head all over again". These results, favoring poly J, correspond with the similar H-shaped design preferences advocated by Nakade, Kamada, Ueno, Kume and Sakaguchi [30]. The advantages of the H-shaped gear shifter were described as follows:

- When drivers select the desired positon, there is only one operation target. The operation start position and operation method are always the same. Therefore, the driver can operate the shift lever without being aware of the current shift position. The shift operation destination is always a dead-end, and there is no need to stop the shift lever in the middle of a shift. This means the driver can make a shift without concern of moving the shift lever too far. This helps to prevent mis-shifts. [30]

The results partly confirm Hypothesis 3; i.e., there was a difference in efficiency and ease of use between gear shifters depending on visual feedback. There was no main effect in terms of efficiency or ease of use depending on whether visual feedback was present or not. This might have been due to the experimental set-up. All participants began with the visual feedback, after which they continued with the no visual feedback condition. Thus, the result might have been due to a training effect. However, an interaction effect occurred between shifter type, complexity and visual feedback; that is, the mono I/J at the highest complexity level and with no visual feedback was clearly less efficient than any other experimental combination. This should be notable since this complexity level is standard in modern cars. Some of the errors that have been reported by US drivers concern the electronic displays malfunctioning and failing to show the active gear mode. As stated by earlier work [14–17], haptics can relieve the visual load. In situations when displays fail, the haptic feeling of the shifter position could help reassure the driver about the actual gear state, reducing the need to direct their eyes off the road.

A monostable gear shifter design is predominantly used in modern cars, providing ample design opportunities and functionality [3,4]. Given this development, from a traffic safety point of view, it is highly important that the implementation of the technology integrates a user perspective. The NHTSA reports, together with our results concerning the difference in efficiency and subjective ease of use

between monostable and polystable joystick gear shifters, indicate that this area of research requires further exploration.

In this paper, we have compared joystick shifters only. Taking into account the emerging diversity among monostable shifter designs, there seems to be a need for increased knowledge vis-à-vis traffic safety and user perspective seems essential.

*Limitations*

There are some limitations with this study which are important to consider in future studies. This study focused solely on the performance of the gear shifters and the task of moving the gear shifter to different positions according to frequent instructions. The gear shifters used in this study were not connected to the driving simulator; i.e., if the participant changed gear to "reverse", the car would not reverse accordingly in the simulator screen. The participants did not use foot pedals or maneuver any other device as is common in a car. Since this study was the first of this kind, it is to be regarded as an exploratory study setting a possible foundation and direction for future studies. The intention was not to create a driving scenario with full ecological validity, since there were limitations with the experimental equipment; the intention was to capture the handling of the gear shifter in a controlled environment with the same high level of distraction for all the participants, such as the LCT task and the frequent gear shifting. This was done in order to mimic a situation in which the drivers have limited time to understand how to use the gear shifter, such as in a car rental scenario. The gear shifting task in itself normally occurs with the car standing still, which was the case in this study. This was also a very safe way to test different gear shifters; also, it was possible to focus specifically on the gear shifter's performance and indicate how intuitive and easy a gear shifter was to understand and learn to use. A full-scale study, studying other factors such as on road performance, could be conducted in a real passenger car with a connection between the gear shifter and the transmission. This could be done in an indoor simulator or under controlled conditions in a natural driving environment. However, this would require more technically complex equipment than that used for this study. Another aspect that could be considered is the duration of the experiment (up to 90 minutes). This could be a long time for a participant to focus on a task. This could cause fatigue and be cognitively demanding or make the participant lose interest. The performance of the gear shifters could have been affected by this; however, the counterbalanced order of the gear shifter testing was implemented to decrease this impact. Additionally, the limitation of participant age should be considered in future studies. A follow-up study with older adults would be of interest as well.

## 5. Conclusions

This study indicates that monostable joystick gear shifters, especially those with an I/J-shape, can be less efficient and easy to use than polystable joystick gear shifters. The monostable joystick with an I/J-shape with reduced visibility and a standard complexity level performed the most poorly of all the tested gear shifter types. The inferiorities of the monostable joystick with an I/J-shape were likely due to the lack of visibility of the gear shifter state. However, the visibility in itself did not seem to be the main issue; rather, the lack of haptic feedback was the most crucial problem. A possibly enhancing factor for all of the gear shifters was the increase in haptic feedback in the shifter pattern. A toggle function could presumably slightly enhance a monostable joystick with an I/J-shape; however, a polystable joystick with a vertical line or I-shape could improve both of the monostable joysticks by providing the distinct feedback of dedicated positions. Furthermore, a polystable joystick with a vertical line but a divided pattern by a J-shape could improve the efficiency and ease of use of the polystable joystick with an I-shape. The results do indicate that additional haptic feedback in the shifter pattern, such as dedicated positions for each gear state, as well as the sectioning of the shifter pattern, would enhance the efficiency and ease of use of the shifter.

**Author Contributions:** Conceptualization, S.L.B. and C.G.; Data curation, S.L.B. and C.G.; Formal analysis, S.L.B., C.G. and P.B.; Funding acquisition, C.G.; Investigation, S.L.B. and C.G.; Methodology, S.L.B. and C.G.; Project

administration, C.G.; Resources, S.L.B. and C.G.; Software, S.L.B. and C.G.; Supervision, C.G. and P.B.; Validation, S.L.B. and C.G.; Visualization, S.L.B., C.G. and P.B.; Writing—original draft, S.L.B. and C.G.; Writing—review & editing, S.L.B., C.G. and P.B. All authors have read and agreed to the published version of the manuscript.

**Funding:** This research was funded by Vinnova, Traffic safety and automated vehicles, FFI, with reference number 2012-04617.

**Acknowledgments:** We would like to thank Jon Friström, the team from Kongsberg Automotive: Henrik Nilsson, Andreas Persson and Nicolas Preisig, and Per-Arne Malm and Cecilia Holtelius from Volvo Car Corporation, for making this project possible. This paper will be a part of a thesis work by Sanna Lohilahti Bladfält.

**Conflicts of Interest:** The authors declare no conflict of interest.

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
