# Peer review of "Comparison of Usability between Gear Shifters with Varied Visual and Haptic Patterns and Complexities"

_mti, doi:10.3390/mti4020024_

Round 1
Reviewer 1 Report
This paper studies a really interesting topic of gear shifter design, with an experimental study involved with 36 participants. Although the style of this paper looks more like a project report instead of a research study, but the results of the experiment do reflect many interesting facts of gear shifter usability. I would suggest the authors to further improve the paper based on my following comments:
- The format in the first section, especially the indention, needs to be greatly improved.
- Questions like “why should you pull the shifter when you are going forward according to the 50 years old PRND standard” sounds very weird. No question should be raised directly to the readers of the paper.
- I have actually some questions for the participants selected in the experiment: The average age of all 36 participants are relatively low, which cannot actually reflect the major body of the drivers. Additionally, what does “driving with automatic” mean? If I understand correctly, why are those two participants that have no experience in automatic transmission chosen?
- The procedure stated in section 2.4.2 can be better illustrated with the help of a flowchart or something like that. The current long paragraph explains many good details, but not in a clear way.
- The results shown in section 3 are very interesting. I am wondering whether it cloud be different if the participants test all different types on a real passenger vehicle instead of a Logitech simulator, especially when they conduct the field implementation while they are actually driving.
Author Response
Thanks for the valuable comments. Please see the attachment.

Reviewer 2 Report
The paper presents a systematic empirical investigation of the objective and perceived ease of use of automatic gear shifters, across different shifter types and complexity levels. A simulator study involving 36 participants is presented, and the results are discussed reflecting on previous findings and NHTSA guidelines.
What was good:
The paper is well written.
The structure is very good, making the paper very easy to follow.
Authors have used standardised approaches (e.g. ISO guidelines) and questionnaires (e.g. TAM).
The gear shifter used in the study is of high fidelity and thus contributed to the realistic feel of the setup.
The paper addressed a very important and safety-critical issue that is little studied, providing essential and empirically grounded knowledge that policymakers and car manufacturers can take into account.
What I think authors could improve:
Major issues:
To make the study described in the paper replicable, Section 2.2. should contain more details about the setup (e.g. what was the size of each screen, what was the distance from the participant to the gear shifter, etc.)
The caption of Figure 3 should contain an explanation of what I, J, and T mean.
I think it would make sense to flip the primary/secondary task labels. In automotive user interface studies, driving is the primary task. This way, we can assess the success of the secondary task, based on how well participants drove while performing it. I suggest, authors check this paper for reference:
Metz, B., Schömig, N. and Krüger, H.P., 2011. Attention during visual secondary tasks in driving: Adaptation to the demands of the driving task. Transportation research part F: traffic psychology and behaviour, 14(5), pp.369-380.
The results of LCT need to be reported and discussed concerning the results of using different gear shifters. It would be exciting to know how the different gear shifters impacted the participants’ driving performance and behaviour.
What I (and probably many of my colleagues) would appreciate seeing in this paper is a Limitations section. There are at least two major limitations that I think authors must acknowledge:
- Ecological validity of the results. The participants had to perform several gear shifts between D, R, N, M, and P, but the vehicle was always driving straight, with a constant speed of 60km/h. I recommend discussing this issue and acknowledging that a more ecologically valid driving task might have impacted the results. For example, participants could drive to a specified location (to testing D, N, and M). Upon arrival, they could park reversely (to test R). Finally, they would put the car into P position before completing the task, just like they would do in real life. I cannot imagine none of the participants found it counterintuitive to go into R mode or to tell their car to park while driving straight on a motorway. I appreciate the fact that matching the driving task with the concept that we want to investigate is not always easy. However, I have seen this being tackles in other papers. Perhaps these sources would serve as a good starting point:
Large, D.R., Banks, V., Burnett, G. and Margaritis, N., 2017, September. Putting the joy in driving: investigating the use of a joystick as an alternative to traditional controls within future autonomous vehicles. In Proceedings of the 9th International Conference on Automotive User Interfaces and Interactive Vehicular Applications (pp. 31-39).
Sirkin, D., Martelaro, N., Johns, M. and Ju, W., 2017, May. Toward measurement of situation awareness in autonomous vehicles. In Proceedings of the 2017 CHI Conference on Human Factors in Computing Systems (pp. 405-415).
Politis, I., Brewster, S. and Pollick, F., 2015, April. To beep or not to beep? Comparing abstract versus language-based multimodal driver displays. In Proceedings of the 33rd annual ACM conference on human factors in computing systems (pp. 3971-3980).
- The long duration of the experiment and the fatigue caused by it. One hour is a long time for human-computer interaction studies. Moreover, participants get bored when driving along a straight motorway for a long time (I wonder if this came up in the interview responses). Consequently, I think it makes sense to at least admit that the experiment was long, and participants’ responses and behaviour might be different in a shorter experiment (perhaps in a pure between-participants design, when each participant drives for a much shorter time).
Finally, I would strongly encourage authors to include a Conclusion section. Discussion is fascinating but also very full of different pieces of information. I believe readers would benefit from a conclusion summarising the critical finding or take-away message of the paper. What is the key thing the authors would like to convey? Is it that poly-stable shifters are more user friendly and save? Is visibility of system’s status the key? Or perhaps the easy-to-understand haptic feedback? It would be very interesting to see the authors’ point of view on this. I am convinced that this would improve the quality of the paper too.
Minor issues:
Reference [3] is meant to provide more guidance on mono- and poly-stable gear shifters, but it does not contain these terms. A novice reader might not know whether “Shift-by-Wire” is mono- or poly-stable.
NHTSA is introduced in the second paragraph of the introduction, although the abbreviation has already been used three times before that. I would encourage authors to explain what NHTSA stands for the first time they mention it in the paper.
Not sure if this is intended, but some paragraphs are differently aligned than others (e.g. lines 50-54 and 58-61). This looks like inconsistent formatting to me. Also, the sentence on line 58 does not start with a capital letter.
I encourage authors to proof-read the article. Some grammar does not seem to be correct (e.g. “shift lead to new questions”, “most limitations … disappears”, and “These driving rounds was completed”).
Which Volvo model was mimicked in the study? This needs to be specified in Section 2.2. I imagine that the specifications will vary from model to model (i.e. not all Volvo cars have the same interiors).
It would be good to have a single photo of the entire setup. Figure 2 appears a bit like a puzzle that the reader needs to put together in their mind, to understand how the four Vmix screenshots are related.
It is quite confusing to have a figure on one page and its caption on another page. I would encourage authors to keep both on the same page (perhaps by making the pictures smaller).
Use the term “participant” instead of “subject”.
It is not clear whether the procedure of the experiment was ethically approved and if the participants received a payment for their contribution to the study (i.e. how much?)
Round 2
Reviewer 1 Report
Thanks for the authors to address all of my previous comments. The authors have added a lot of contents into the new version of the manuscript, which is really good. I would like to request the authors to conduct a final round of spelling/format check before proceeding to the next step. For example, the new figure 5 added by the authors seems to have some format issue.
Author Response
Thanks for your valuable comments, we have revised in the highlight manuscript.